

# Synchronizing smart city nodes using Skew Integrated Timestamp (SIT)

Muhammad Usman Hashmi[1], Muntazir Hussain[2], Asghar Ali Shah[1], Muhammad Babar[3] and Basit Qureshi[4]

[1] Department of Computer Science, Bahria University, Islamabad, Islamabad, Islamabad, Pakistan
[2] Department of Electrical and Computer Engineering, Air University Islamabad, Islamabad, Islamabad, Pakistan
[3] Robotics and Internet of Things Lab, Prince Sultan University (PSU), Riyadh, Riyadh, Saudi Arabia
[4] College of Computer and Information Sciences, Prince Sultan University, Riyadh, Riyadh, Saudi Arabia

## ABSTRACT

Time synchronization among smart city nodes is critical for proper functioning and coordinating various smart city systems and applications. It ensures that different devices and systems in the smart city network are synchronized and all the data generated by these devices is consistent and accurate. Synchronization methods in smart cities use multiple timestamp exchanges for time skew correction. The Skew Integrated Timestamp (SIT) proposed here uses a timestamp, which has time skew calculated from the physical layer and uses just one timestamp to synchronize. The result from the experiment suggests that SIT can be used in place of multiple timestamp exchanges, which saves computational resources and energy.

## INTRODUCTION

A smart city uses state-of-the-art technology for the improvement and enhancement of various urban services and provides a better lifestyle to the residents. Smart city technology typically includes the Internet of Things (IoT), which enables devices and systems to collect and share data in real-time. In smart cities advanced analytics and machine learning algorithms helps to optimize decision-making (*Hassan et al., 2021*; *Arasteh et al., 2016*; *Ghazal et al., 2023*; *Ghazal et al., 2021*; *Koubaa et al., 2020*; *Babar et al., 2021*; *Umer et al., 2023*). IoT forms the backbone of many smart city initiatives by connecting various devices, sensors, and systems to gather data, facilitate communication, and enable intelligent decision-making.

The Internet of Things (IoT) uses smart devices to collect data from various sources in a smart city, including traffic sensors, environmental sensors, smart meters, surveillance cameras, *etc.*, (*Gaur et al., 2015*; *Aditya et al., 2023*; *Bellini, Nesi & Pantaleo, 2022*; *Poongodi et al., 2021*; *Babar, Arif & Irfan, 2019*; *Alshathri et al., 2023*; *Pranto et al., 2021*). These smart devices generate data communicated over the network, which needs to be processed and analyzed in real-time. The IoT technology integrates various city systems, *e.g.*, energy, water, transportation, and waste management. It connects people to infrastructure and even vehicles. Data collected from these various IoT devices can be analyzed by city

Corresponding author
Muntazir Hussain,
muntazir_hussain14@yahoo.com

administrators, who can make guided decisions to optimize the allocation of resources. In this way, city administrators can improve their operational efficiency. The use of applications and services by smart city citizens allows them to remain updated in real-time. Their participation through their response to the community is increased and encouraged. Citizens have updated information about public services, and they can actively contribute to development and management by providing feedback. Smart cities aim to enhance the quality of life in various ways, *e.g.*, smart parks can sense air quality and inform the citizens. Hence, IoT is a main technology that forms the basis of smart cities as IoT enables data collection and analysis to create more sustainable and livable urban environments (*Bellini, Nesi & Pantaleo, 2022*; *Cvar et al., 2020*; *Hassan et al., 2021*).

IoT-based smart city offers various benefits, for instance, improved efficiency and quality of life, better resource management, economic development, and increased sustainability. To achieve these benefits coordination between IoT nodes is required (*Arasteh et al., 2016*; *Gaur et al., 2015*; *Ghazal et al., 2023*; *Ali et al., 2023*; *Cvar et al., 2020*; *Hassan et al., 2021*; *Saba, Sadad & Hong, 2022*). IoT nodes are smart devices typically equipped with sensing, communication capabilities, and computing power to collect and exchange data with other smart devices within a smart city environment. When these IoT nodes are synchronized in time, they have the capability to timestamp events, providing a chronological reference for a more comprehensive understanding of the overall scenario. Synchronized nodes work together in a coordinated and efficient manner (*Yiğitler, Badihi & Jäntti, 2020*; *Tirado-Andrés, Rozas & Araujo, 2019*; *Hashmi et al., 2023*; *Hashmi et al., 2021a*; *Hashmi et al., 2021b*). Time synchronization enhances predictive maintenance, disaster response, and real-time decision-making, creating a more sustainable and efficient urban environment. Time synchronization is crucial for effectively coordinating IoT nodes in a smart city (*Fan et al., 2019*; *Skiadopoulos et al., 2019*; *Fan et al., 2022*).

There are several solutions to synchronization problem (*Yiğitler, Badihi & Jäntti, 2020*; *Tirado-Andrés, Rozas & Araujo, 2019*; *Yoo, Park & Lee, 2020*; *Skiadopoulos et al., 2019*), including the Network Time Protocol (NTP), Precision Time Protocol (PTP), Global Navigation Satellite Systems (GNSS), and the Distributed Time Synchronization Protocol (DTSP). Based on the available information and the author's understanding, every method uses several exchanges of timestamps for the estimation of clock frequency offset (skew) among synchronizing nodes. The SIT method proposed here requires one timestamp and incorporates skew correction information with a time phase value. Reducing timestamps increases the energy efficiency of smart cities, which can bring several benefits, including cost savings, environmental benefits, increased reliability and comfort, enhanced resilience, and economic growth.

The organization of the article is made in six sections. 'Literature Review' elaborates on the literature available for time synchronization among smart city nodes. 'Methodology' describes the proposed Skew Integrated Timestamp (SIT) synchronization methodology. 'Mathematical Analysis' provides mathematical analysis and details. 'Experimentation and Results' discusses the experimental details and results, showing that skew can be corrected with a single timestamp by using the SIT method. In light of the results, 'Conclusions' concludes that the proposed SIT method can lead to efficient time synchronization.

## LITERATURE REVIEW

A synchronization error of a clock occurs when the clock's time is not aligned with the actual time. In a smart city network, every node's clock can drift away in time due to various factors, including clock skew, network delays or even temperature variations. Overall network efficiency is reduced because of non-synchronized nodes resulting in incorrect timing of events. Synchronization errors can be prevented by periodically synchronizing the whole network using any reliable time synchronization method. Various time synchronization mechanisms are in place for smart cities. The Network Time Protocol (NTP) (*Mills et al., 2010*), Precision Time Protocol (PTP) (*Eidson, Fischer & White, 2002*), Flooding Time Synchronization Protocol (FTSP) (*Maróti et al., 2004*), Blockchain-based time synchronization (*Fan et al., 2019*; *Fan et al., 2022*) and the Distributed Time Synchronization Protocol (DTSP) are few state of the art methods.

Network Time Protocol (NTP) is a widely used protocol for time synchronization of nodes in many networks, including smart city network (*Mills et al., 2010*; *Hou et al., 2022*; *Sommars, 2017*). NTP creates a hierarchical structure of nodes for synchronization. A tree-like structure of smart city nodes is created. The highest-level nodes have master clocks (nodes with external time references such as atomic clocks or GPS). Lower-level nodes need to synchronize with the higher-level nodes. With NTP, millisecond accuracy can be achieved by using a combination of intersection and clustering algorithms, internal clocks, and master clocks.

The Precision Time Protocol (PTP), also known as IEEE 1588, uses a peer-to-peer architecture to achieve microsecond accuracy (*Eidson, Fischer & White, 2002*; *Watt et al., 2015*; *Resner, Fröhlich & Wanner, 2016*). It is considered a more advanced protocol that uses timestamps and delay measurement techniques. PTP architecture is master–slave architecture: the master clock sends time synchronization messages to the slave clocks. Upon receiving of synchronization messages, the slave clocks synchronize their time by adjusting the clock phase and skew. PTPv1 and PTPv2 are two versions of PTP. PTPv2 is more promising with better accuracy if network infrastructure and configuration are designed to minimize network delays.

The Flooding Time Synchronization Protocol (FTSP) is a widely deployed protocol in wireless sensor networks (WSNs) and also has its application in smart cities (*Maróti et al., 2004*; *Gheorghe, Rughinis & Tapus, 2010*). FTSP is a distributed protocol central time server or clock. Flooding of time synchronization messages is done periodically among neighboring sensor nodes. Each node broadcasts its local time, and neighbors adjust their clocks based on the received timestamp. The protocol employs techniques such as timestamp re-synchronization and clock rate estimation to mitigate clock skew and achieve synchronization accuracy.

Blockchain-based time synchronization refers to utilizing blockchain technology to establish a trusted and decentralized mechanism for synchronizing time among various devices and systems in a smart city environment (*Fan et al., 2019*; *Fan et al., 2022*). Traditional time synchronization methods, such as NTP, PTP, and FTSP, rely on centralized time servers or hierarchical structures, which may introduce vulnerabilities

or single points of failure. Instead of relying on a central time server, blockchain-based time synchronization distributes the time authority across multiple nodes in the network. These nodes reach a consensus on the accurate time through a consensus mechanism. Each event or transaction in the smart city network is assigned a timestamp recorded on the blockchain. The timestamp represents the agreed-upon time established by the distributed time authority. This allows for the chronological ordering and verification of events across the network. Once a timestamp is recorded on the blockchain, it becomes immutable and transparent. It cannot be altered retroactively, providing a tamper-proof record of time synchronization. This enhances the integrity and trustworthiness of the time information in the smart city network.

The Global Navigation Satellite System (GNSS) is a constellation of satellites that provide positioning, navigation, and timing information to users on Earth (*Zaidi & Suddle, 2006*). The most well-known GNSS system is the Global Positioning System (GPS) operated by the United States, but there are other systems such as GLONASS (Russia), Galileo (European Union), and BeiDou (China). In the context of smart cities, GNSS plays a crucial role in enabling location-based services, accurate positioning, and synchronization of devices and systems. GNSS/GPS provides highly accurate time signals, which are essential for time synchronization in smart city networks (*Zaidi & Suddle, 2006*; *Dana, 1997*). Devices can receive GNSS signals to synchronize their internal clocks, ensuring coordination and accurate timing in applications. However, GNSS signals can be affected by environmental factors like tall buildings, dense urban environments, or signal interference. Advancements in GNSS technology, including multi-constellation support and enhanced signal processing techniques, are addressing these challenges and improving accuracy in urban areas.

Any of the above-mentioned methods can be used for time synchronization in smart cities. However, they all require several exchanges of timestamps for the estimation of skew. The proposed SIT method uses one timestamp to estimate skew.

The main contributions of this article are:

1. In contrast to the existing methods (*Maróti et al., 2004*; *Gheorghe, Rughinis & Tapus, 2010*; *Zaidi & Suddle, 2006*; *Dana, 1997*; *Fan et al., 2019*; *Fan et al., 2022*), the proposed Skew Integrated Timestamp technique is more efficient. Because the existing methods, for instance, (*Maróti et al., 2004*; *Gheorghe, Rughinis & Tapus, 2010*; *Zaidi & Suddle, 2006*; *Dana, 1997*; *Fan et al., 2019*; *Fan et al., 2022*), require several exchanges of timestamps for the estimation of skew. Whereas the proposed SIT method uses one timestamp to estimate skew.

2. Energy efficiency is improved for the reason that multiple timestamps are no longer required to extract the clock skew of the application layer clock.

3. SIT method is experimentally implemented to show that skew correction can be done using one timestamp. TMS320C6713 (Texas Instruments, Dallas, TX, USA) are DSP Starter Kits (DSKs) and are used as nodes in the experiment transmitting node A (DSK A) and receiving node B (DSK B).

The following section shows the details of the proposed methodology.

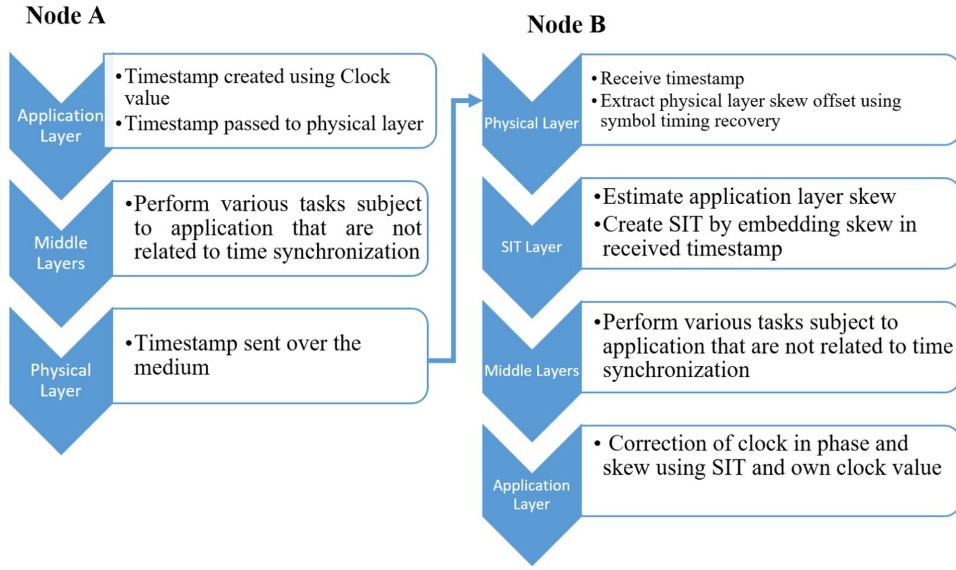

**Figure 1** Node synchronization using SIT.

## METHODOLOGY

Application layer clocks typically function as timers, tracking the number of oscillations of a crystal and utilizing two registers to determine the equivalence between the oscillations and a single tick of clock running at application layer (*Sundararaman, Buy & Kshemkalyani, 2005*). Additionally, the clock running at the application layer and the clock responsible for symbol timing synchronization at the physical layer of the same node also rely on counting the oscillations of the quartz crystal of the hardware clock. To elaborate on how the proposed Skew Integrated Timestamp (SIT) method works, consider that two nodes (Node A and B) need time synchronization as shown in Fig. 1. Node A acts as a node with accurate time information and node B needs to be aligned with node A by correcting its time phase and skew.

### Steps in node synchronization using SIT

The steps of the method are as follows.

1. Node A will create a timestamp using its application layer master clock value and transfer it to receiving node B.
2. Node B will receive the timestamp and implement symbol timing recovery at its physical layer for the estimation of the skew.
3. This computed skew of the physical layer is used to estimate the application layer skew. **Since both the physical and application layer clocks are derived from the hardware clock within the same node, it is essential for their skew to be identical.**
4. Now estimated application layer skew is embedded within the received timestamp, creating the Skew Integrated Timestamp (SIT).

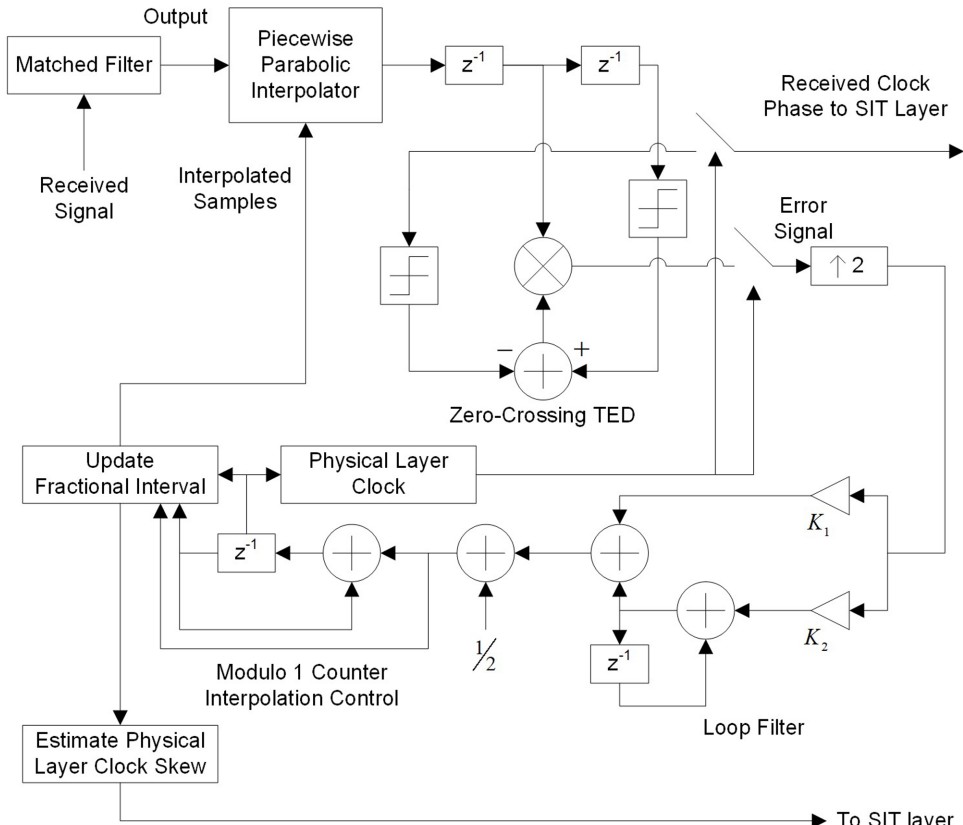

**Figure 2** Symbol timing recovery for skew estimate.

5. SIT will now be used by the application layer of node B to correct its phase and skew offset.

6. Node A and Node B get synchronized in time by using one timestamp by inserting SIT method at receiving node B.

Figure 1 shows the process of synchronization among two nodes using SIT.

To further elaborate the steps involved in receiving node B symbol timing recovery are discussed next. Detailed discussions of various synchronization systems at the physical layer can be found in the works of *Rice (2008)* and *Meyr, Moeneclaey & Fechtel (1998)*. These synchronization systems have different modulation schemes, implementation of various interpolators, special types of timing error detectors, multiple loop filters, and diverse interpolation controls. Figure 2 presents one such system, modified for SIT.

## Physical layer configuration for skew estimate

The physical layer configuration of the receiving node is as follows,

- Modulation scheme: Binary Pulse Amplitude Modulation (PAM).
- Interpolator: Piecewise Parabolic.
- Timing Error Detector: Zero Crossing Timing Error Detector (TED).
- Loop Filter: Proportional plus Integrator.

- Interpolation Control: Mod-1 Counter.

This system effectively traces and amends for phase and frequency errors (skew) in the physical layer clock.

## MATHEMATICAL ANALYSIS

The received signal at node B is passed to matched filter which processes the signal. The processed outcome $x(nT)$ is fed to piecewise parabolic interpolator for p-th interpolant and can be written as,

$$x((m(p)+\mu(p))T) = \left(\frac{\mu(p)^3}{6} - \frac{\mu(p)}{6}\right)x((m(p)+2)T) - \left(\frac{\mu(p)^3}{2} - \frac{\mu(p)^2}{2} - \mu(p)\right)$$
$$x((m(p)+1)T) + \left(\frac{\mu(p)^3}{2} - \mu(p)^2 - \frac{\mu(p)}{2} + 1\right)x(m(p)T) \qquad (1)$$
$$- \left(\frac{\mu(p)^3}{6} - \frac{\mu(p)^2}{2} + \frac{\mu(p)}{3}\right)x((m(p)-1)T).$$

Here, $p$ is the p-th interpolant which is the new sample which reduces the error margin each time. The value is between $+1$ and $-1$. $m(p)$ represents the basepoint index, and $\mu(p)$ is the fractional change. This is the output of the Mod-1 counter which will be used by the interpolator for the computation of the next interpolant. Zero Crossing TED attempts to adopt the signal of interpolator for tracking of error and when perfectly aligned, it produces a zero error (*Rice, 2008*). The error signal $e(p)$ in p-th interpolant can be determined from TED by Eq. (2).

$$e(p) = x\left((p-1/2)T_s+\tau\right)\left[sgnx\left((p-1)T_s+\tau\right) - sgnx(pT_s+\tau)\right], \qquad (2)$$

where *sgnx* represents the sign of x. The loop filter, with constant coefficients $K_1$ and $K_2$, generates an output that is passed to the Mod-1 counter to estimate $\mu(p)$ and $m(p)$. $K_1$ and $K_2$ is carefully calculated using sample time $T$, symbol time $T_s$ and noise bandwidth (*Rice, 2008*). The performance of the synchronization system in terms of tracking and acquisition time relies on these loop filter parameters. The Mod-1 counter takes advantage of the signal from loop filter $\upsilon(n)$, to determine $m(p)$ and $\mu(p)$. The values of Mod-1 counter $\eta$ are computed as,

$$\eta(n+1) = (\eta(n) - T/T_s - \upsilon(n)) \qquad (3)$$

when $\eta$ underflows, $n$ is equal to basepoint index $m(p)$. To find the fractional interval at the basepoint index, Eq. (4) is used,

$$\mu(m(p)) = \eta(m(p))/(T/T_s + \upsilon(m(p))). \qquad (4)$$

The values of $\mu(m(p))$ and $m(p)$ are then employed to evaluate the value of the upcoming interpolant.

By employing the aforementioned iterative mechanism, the variations of $\mu(p)$ can be used to estimate skew. Various mechanisms for computing the skew are described in the works of *Rice (2008)*, *Mengali & D'Andrea (1997)* and *Meyr, Moeneclaey & Fechtel (1998)*.

*Rice (2008)* proposes that skew $\xi_P$ is the slope of the fractional change that can be computed using any method including least square. Now the new sampling rate $f_s$ can be written with the help of symbol rate $f_y$ as Eq. (5), to achieve symbol timing recovery at the physical layer.

$$f_s = (2 + \xi)f_y. \tag{5}$$

A simple calculation of skew can be done by making a line between the first point k = 1 and the last point $k = p$ of $\mu(k)$ as given by using Eq. (6).

$$\xi_P = \frac{\mu(p) - \mu(1)}{p - 1}. \tag{6}$$

SIT layer is next responsible for mapping physical layer clock skew $\xi_P$ to application layer clock skew $\xi_A$. Keeping the fact in mind that the hardware clock is driving both clocks of the physical and application layer. This means that $\xi_P$ is actually the skew of hardware clock $\xi_H$ and $\xi_H$ must be the same as of $\xi_A$ can be related by Eq. (7) and as shown by the experimentation results in experiment section.

$$\xi_H = \xi_P = \xi_A. \tag{7}$$

Now $\xi_A$ is appended in the timestamp received with the sender's clock phase value $\phi(S)$ and passed to the application layer of receiving node B. This appended timestamp is the Skew Integrated Timestamp (SIT). This SIT is now used at the application layer for rectifying application layer clock skew using $\xi_A$ and also phase such that the clock value of the node B $\phi(R)$ is available at the application layer and can be used to compute phase offset as $\phi(R) - \phi(S)$ and synchronize in phase along with skew. Now Node A and Node B clocks are synchronized using one timestamp by implementing SIT at Node B.

## EXPERIMENTATION AND RESULTS

An experimentation setup is prepared to prove that the physical layer clock skew and application layer clock skew are the same since both are derived from the same hardware clock within the node. Once the fact is established, the SIT method can be used to synchronize the two nodes using one timestamp.

### Experimental objectives

The objectives of the experiment are as follows:

1. Verify equivalence of physical and application layer clock skew: Demonstrate that the physical and application layers clock skew are identical, since both the physical and application layer clocks are derived from the hardware clock within the same node.
2. Establish the basis for synchronization using the SIT method: Once the equivalence of clock skew is achieved, then the SIT method is employed to synchronize two nodes using a single timestamp.
3. Utilize TMS320C6713 DSP Starter Kits: Implement the experiment using TMS320C6713 DSP Starter Kits (DSKs) (Texas Instruments, Dallas, TX, USA) as the transmitting node A (DSK A) and receiving node B (DSK B).

4. Include physical and application layer synchronization systems: Set up both a physical layer synchronization system and an application layer synchronization system to ensure a comprehensive evaluation.

5. Designate DSK A clock as the reference: Consider the clock of DSK A as having accurate clock information, and aim to synchronize DSK B with the clock information from DSK A.

6. Configure DSKs for implementation of communication system: Configure DSK A as transmitter and DSK B as receiver in order to implement comunication system.
   These objectives collectively aim to validate the synchronization approach.

## Experimental configuration

The physical layer configuration of used DSKs are as follows:

1. Configuration of physical layer of DSK A for transmission: Configuration of the physical layer with specific parameters required for communication system are as follows:

   - Modulation: Binary PAM
   - Symbols: $+1, -1$
   - Transmitted symbols: 10,000
   - Symbol rate: 4,000 symbols/s
   - Sampling rate: 16,000 samples/s
   - Samples/Symbol: 4
   - Samples: 40,000
   - Transmission time: 2.5 s
   - Upsampling factor: 4
   - Pulse Shaping filter: Square Root Raised Cosine (SRRC) with 50% excess bandwidth.

2. Configuration of physical layer of DSK B for reception: At the receiving end (DSK B), initiate the physical layer processing using the symbol timing recovery system, mirroring the configuration outlined in the experiment.Additional physical layer configuration and details are as follows:

   - Received symbols undergo matched filtering with SRRC.
   - Downsample to two samples/symbol.
   - Apply a piecewise parabolic interpolator to downsampled symbols.
   - Pass interpolated symbols through a Zero Crossing TED.
   - Use the error signal from TED in the proportional plus integrator loop filter.
   - Utilize the loop filter output for Mod-1 counter interpolation control.
   - The loop filter parameters required for efficient acquisition time and tracking time is set using noise bandwidth $= 1/\sqrt{2}$, $K_p = 2.7$, $K_0 = -1$, and $N = 2$. Where $K_p$ and $K_0$ are used for loop gain calculations $K_1$ and $K_2$.

## Experiment methodology

System presented in Fig. 2 is implemented using the configuration discussed above. The error signal and fractional interval are obtained and depicted in Fig. 3. The fractional

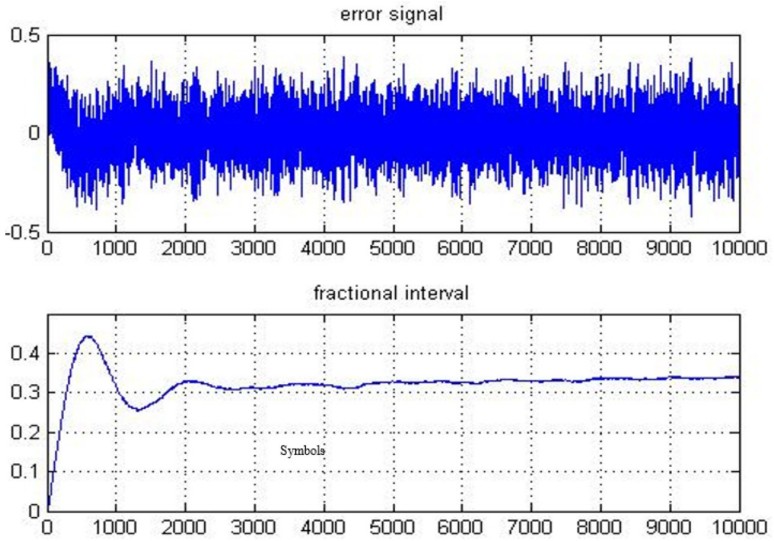

**Figure 3   Error signal and fractional interval examined for received symbols.**

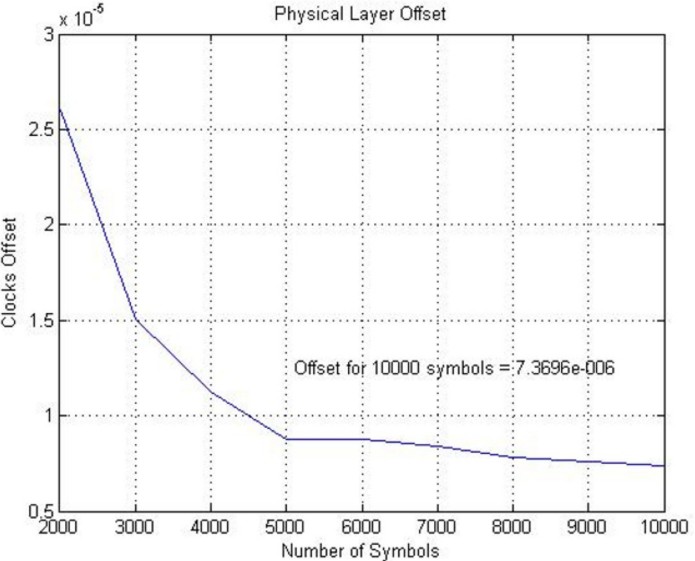

**Figure 4   Physical layer skew computed from fractional interval.**

interval slope is the skew among DSK A and B clocks. The least squares (LS) estimation approach is employed to determine the slope. The accuracy of the frequency offset estimate improves with an increasing number of transmitted symbols, as demonstrated in Fig. 4.

On application of LS on the fractional intervals with various transmitted symbols, ranging from 2,000 to 10,000, gives the skew estimate using a specified number of symbols. The results show that the estimate of the skew becomes more accurate with a larger number

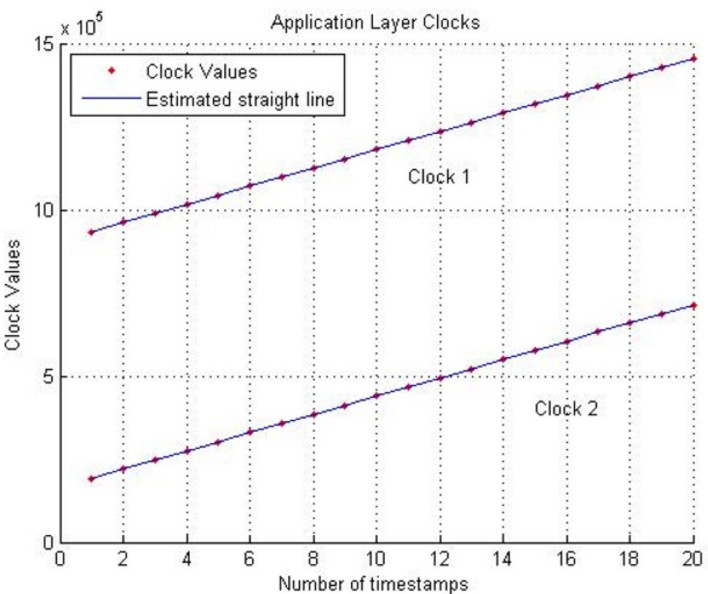

**Figure 5 Application layer clocks.** Clock 1:DSK A and Clock 2:DSK B.

of transmitted symbols. Over 10,000 symbols that were originally transmitted from DSK A, the estimated physical layer clock skew is determined to be 7.3696 ppm.

For the skew estimation of the application layer clock of the same DSKs that are used above, multiple timestamps are exchanged. DSK A transmits twenty timestamps over the same transmission time *i.e.,* 2.5 s. DSK B records their respective clock values upon reception of timestamps. This synchronization mechanism at the application layer uses LS estimation on these timestamps for the computation of the skew. In Fig. 5, "Clock 1" belongs to DSK A, and "Clock 2" belongs to DSK B.

LS estimation method is applied to values of Clock 1 and Clock 2. The slope of the line representing Clock 1 and Clock 2 is the frequency of DSK A's clock and DSK B's clock. The offset among slopes represents the skew (frequency offset) between Clock 1 and Clock 2. Figure 6 illustrates the application layer skew computed against a different number of timestamps. It is observed that increasing the number of timestamps enhances the accuracy of the computed skew. For twenty timestamps, the skew is determined to be 7.3637 ppm, which is approximately identical to the physical layer clock skew of 7.3696 ppm.

Figure 7 provides a comparison between the skew (frequency offset) at the application and physical layer for 2.5 s of transmission time. Twenty transmitted timestamps correspond to 2.5 s of transmission time and with 4,000 symbols/s, the transmission of 10,000 symbols also corresponds to 2.5 s of physical layer communication. From Fig. 7, it can be observed that the clock skew at both layers converges after one second of transmission, equivalent to the transmission of 4,000 symbols. This demonstrates that the skew is the same at both layers as expected.

SIT exploits this fact established from the above experiment and uses one timestamp, instead of multiple. The process of synchronization among two nodes using SIT method is

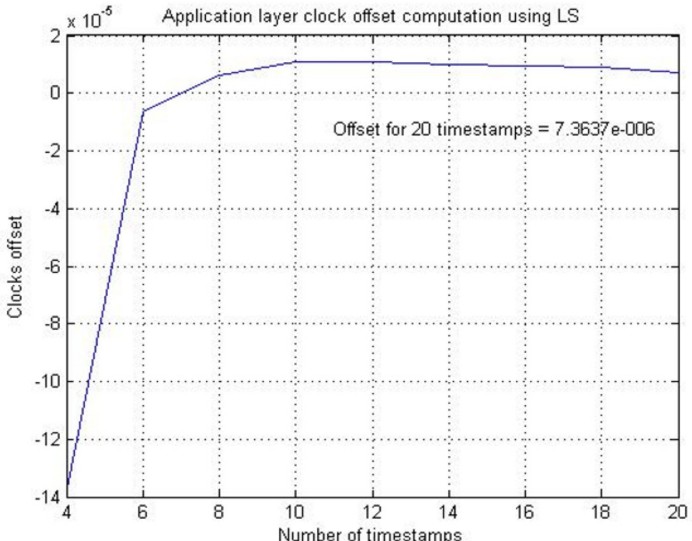

**Figure 6  Application layer skew computed using various timestamps.**

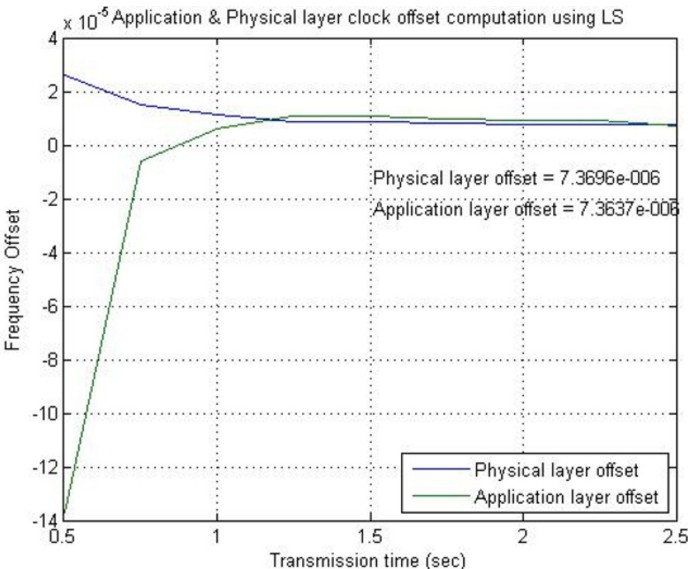

**Figure 7  Skew (frequency offset) comparison of physical layer clock and application layer clock.**

explained in Figs. 1 and 2. The following procedure is followed in SIT to synchronize DSK A and DSK B:

- DSK A creates a timestamp with its application layer clock value 23.
- Timestamp is converted to symbols and transmitted over the medium to DSK B.
- Physical layer of Node B receives the symbols and passes them from the system model shown in Fig. 2.

- Physical layer skew of node B is computed to be 7.3696 ppm and passed to SIT layer and symbols are converted back to received timestamp value 23.
- SIT layer equates the application layer clock skew with the physical layer clock skew *i.e.,* 7.3696 ppm.
- SIT layer creates SIT by appending the received timestamp value with application layer skew. SIT has two values 23 and 7.3696 ppm and passes them to the application layer.
- Node B application layer clock value is 59 which is used to compute clock phase offset as 59−23 = 36. By knowing the phase offset Node A and Node B synchronizes in the clock phase.
- Node B uses SIT (skew 7.3696 ppm) to correct application layer clock skew and hence synchronize in frequency.
- By using SIT method, both nodes are synchronized in clock phase and frequency by using just one-timestamp.

Number of symbols required to estimate skew at the application layer have a direct relation with accuracy. If these symbols are below a certain number then precise synchronization may not be achieved which may be a requirement in some areas of smart city applications.

## CONCLUSIONS

The Skew Integrated Timestamp (SIT) method presented in this article offers a highly promising approach to enhance the efficiency of smart city networks. Through experimentation and demonstrated results, it has been shown that the SIT approach significantly reduces the resources needed for distributing multiple timestamps. By implementing SIT at the receiving node, a single timestamp is sufficient to synchronize two nodes in time, resulting in computational and energy savings. SIT proves to be particularly advantageous for nodes with limited battery power in smart cities, as it can be applied to all synchronization-dependent nodes. Its utilization among all nodes can exponentially improve the overall energy efficiency of smart city networks. In comparison to existing methods, the proposed SIT technique stands out for its efficiency. While existing methods require multiple exchanges of timestamps to estimate skew, the SIT method accomplishes this with just one timestamp. This streamlined process eliminates the need for multiple timestamps to extract the clock skew of the application layer clock, further enhancing energy efficiency. The experimental implementation of the SIT method confirms that skew correction can indeed be achieved using a single timestamp. Overall, the findings suggest that the SIT method holds great promise for optimizing the efficiency and performance of smart city networks, making it a compelling solution for synchronization challenges.

### Funding

Prince Sultan University paid the Article Processing Charges (APC) for this publication. The funders had no role in study design, data collection and analysis, decision to publish, or preparation of the manuscript.

### Grant Disclosures

The following grant information was disclosed by the authors:
Prince Sultan University paid the Article Processing Charges (APC).

### Competing Interests

The authors declare there are no competing interests.

### Author Contributions

- Muhammad Usman Hashmi conceived and designed the experiments, prepared figures and/or tables, and approved the final draft.
- Muntazir Hussain conceived and designed the experiments, performed the experiments, prepared figures and/or tables, and approved the final draft.
- Asghar Ali Shah analyzed the data, authored or reviewed drafts of the article, and approved the final draft.
- Muhammad Babar performed the computation work, authored or reviewed drafts of the article, and approved the final draft.
- Basit Qureshi analyzed the data, performed the computation work, authored or reviewed drafts of the article, and approved the final draft.

### Data Availability

The raw data and code are available in the Supplemental Files.

### Supplemental Information

Supplemental information for this article can be found online at http://dx.doi.org/10.7717/peerj-cs.2049#supplemental-information.

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
