# Peer review of "Synchronizing smart city nodes using Skew Integrated Timestamp (SIT)"

_PeerJ Computer Science, doi:10.7717/peerj-cs.2049_

## Round 0.1 · original submission · Major Revisions

· Academic Editor

Major Revisions

Upon reviewing feedback from qualified reviewers, it is evident that your manuscript necessitates significant revisions. Make major changes based on the comments provided. Ensure to address these revisions thoroughly before resubmitting your manuscript.

·

Basic reporting

Here are some areas where the article could be improved:

Clarity and Organization: The methodology and mathematical analysis section is hard to follow due to the dense text and lack of clear organization. It would be helpful to break down the process into sub-sections or use bullet points to make the steps more distinct and easier to understand.

Language and Grammar: Several sentences could be rephrased for clarity. For instance, "Keeping the fact in mind that skew at the physical layer and application layer has to be the same because both clocks are derived by hardware clock within the same node" could be rewritten more clearly as "Since both the physical and application layer clocks are derived from the hardware clock within the same node, it's essential for their skew to be identical."

Conciseness: Some sentences are overly long and could be broken down into smaller, more digestible parts to enhance readability.

Incorporating these improvements will make the article more accessible and enhance its quality.

Experimental design

The article's experimental design has several areas where it can be improved:

Clear Experimental Objectives: The objectives of the experiment are not explicitly stated. It would be beneficial to clearly define the research questions or objectives that the experiment aims to address. For example, what specific aspects of physical layer and application layer synchronization are being investigated?

Methodology Description: The article lacks a detailed description of the methodology used in the experiment. It's important to provide step-by-step instructions and explain how the experiment was conducted, including the setup, equipment, and data collection procedures.

Validity of the findings

It is suggested that the article should provide a more comprehensive and well-structured description of the experimental process, along with clear explanations of the data presented.

It also notes the absence of clear objectives and detailed methodology. The lack of statistical analysis and data interpretation is emphasized, and there is a call for a more consistent and concise presentation. Additionally, the need to discuss the experiment's limitations is also highlighted.

Reviewers emphasize the importance of addressing these issues to enhance the validity of the article's findings and to make it more informative and rigorous.

Additional comments

The paper can be accepted after major modification/ revision.

Cite this review as

·

Basic reporting

Line (28) missing reference
Line 49 The authors use “IOT nodes” without a clear definition of it.
The font in Figure 1 is not clear.
Line 50 Please review “When these IOT nodes are time synchronized, they can tag the events with time for a better understanding of the bigger picture.”
Line 57: “To the best of the author’s knowledge”,
Enhance the resolution of Figures 2,3,4,5,6,7.

Experimental design

THE EXPERIMENTATION AND RESULTS part needs to be re-described, as, from my point of view, the way it was presented is incomprehensible and needs further clarification. I also believe that providing this part with a drawing may add more clarity about the experiment setup.


Line 52: “How can Time synchronization enhance predictive maintenance”
Line 58: “Every method uses several exchanges of timestamps…”, proof.
What is Equation 1?
Explain more variable (p) used in “METHODOLOGY AND MATHEMATICAL ANALYSIS” and show the range of its values.
Define e(p) in equation 2
what is (n) in equation 3, and how is it calculated?

Validity of the findings

Lines (225 – 234) What are the justifications for the used values in the Physical layer configuration details?
In Figure 6 the computed skew belongs to which clock!

Cite this review as

---

## Round 0.2 · accepted · Accept

· Academic Editor

Accept

Please follow the next steps from the production team.

·

Basic reporting

I thank the researchers for using the given comments and for adding further additions that enhanced the manuscript.
After reviewing the manuscript, I found that they had modified the research for the better.

Experimental design

ok

Validity of the findings

ok

Additional comments

ok

Cite this review as